# Mood Stabilizers of First and Second Generation

**DOI:** 10.3390/brainsci13050741

**Published:** 2023-04-29

**Authors:** Janusz K. Rybakowski

**Affiliations:** Department of Adult Psychiatry, Poznan University of Medical Sciences, 61-701 Poznan, Poland; janusz.rybakowski@gmail.com

**Keywords:** antipsychotic drugs, first generation, second generation, third generation, bipolar mood disorder, mania, bipolar depression, maintenance treatment

## Abstract

The topic of this narrative review is mood stabilizers. First, the author’s definition of mood-stabilizing drugs is provided. Second, mood-stabilizing drugs meeting this definition that have been employed until now are described. They can be classified into two generations based on the chronology of their introduction into the psychiatric armamentarium. First-generation mood stabilizers (FGMSs), such as lithium, valproates, and carbamazepine, were introduced in the 1960s and 1970s. Second-generation mood stabilizers (SGMSs) started in 1995, with a discovery of the mood-stabilizing properties of clozapine. The SGMSs include atypical antipsychotics, such as clozapine, olanzapine, quetiapine, aripiprazole, and risperidone, as well as a new anticonvulsant drug, lamotrigine. Recently, as a candidate for SGMSs, a novel antipsychotic, lurasidone, has been suggested. Several other atypical antipsychotics, anticonvulsants, and memantine showed some usefulness in the treatment and prophylaxis of bipolar disorder; however, they do not fully meet the author’s criteria for mood stabilizers. The article presents clinical experiences with mood stabilizers of the first and second generations and with “insufficient” ones. Further, current suggestions for their use in preventing recurrences of bipolar mood disorder are provided.

## 1. Introduction

For this narrative review, two preliminary assumptions should be made. The first is a definition of a mood stabilizer (MS). I propose a definition that includes the impact of the drug on bipolar disorder, both in acute and long-term treatment. The first criterion is a therapeutic effect on manic and/or depressive symptoms during an acute episode. The second indicator, which seems the most essential, requires the prevention of manic and/or depressive recurrences. The drug should be given as monotherapy, and the administration’s duration should be at least one year. According to the third necessary feature, the drug should not produce or exacerbate manic or depressive episodes or mixed states. Such a criterion is, therefore, not met by typical antipsychotic medications, which may cause depression and by antidepressants that can induce mania [1].

The definition including these three criteria established a basis of my suggestion in 2007 for a classification of MSs based on the chronology of their inauguration in psychiatry for such purposes [2]. The first generation of mood stabilizers (FGMSs), such as lithium, valproates, and carbamazepine, began in the 1960s and 1970s. It was not until the mid-1990s that the mood-stabilizing feature of clozapine was unearthed [3]. This discovery began the second generation of mood stabilizers (SGMSs). This chronological distinction made the second assumption.

In the article, clinical experiences with FGMSs and SGMSs will be presented as well as suggestions for their use in bipolar mood disorder. Drugs with “insufficient” criteria for MSs will be also mentioned.

## 2. First Generation of Mood Stabilizers (FGMSs)

### 2.1. Lithium

Sixty years ago, an article appeared, demonstrating, for the first time, the possibility of preventing recurrences in mood disorders by using lithium salts. The author of the paper was a British psychiatrist, Geoffrey Philip Hartigan (1917–1968), known as “Toby”, working in St. Augustine’s Hospital in Chartham Down [4]. Among 45 patients receiving lithium carbonate in this institution over six years, he presented his observations on long-term lithium administration (≥three years) in seven patients with bipolar mood disorders (BDs) and eight patients with recurrent depression. There were no recurrences of the illness in six persons from the first group and six from the second group. This publication appeared 14 years after an article from an Australian psychiatrist, John Cade, showing the therapeutic effect of lithium in manic states [5]. The following year after Hartigan’s paper, a Danish psychiatrist, Poul Christian Baastrup, established the “prophylactic” effect of lithium in 11 BD patients receiving the drug for three years [6].

The pioneering idea from Hartigan was to point out the possibility of the favorable effect of lithium administration on the long-term course of mood disorders, both BD and recurrent depression. He referred to the terms “normothymotics” and “mood-normalizers” proposed by a Danish psychiatrist, Mogens Schou, in an article published in the same issue of the *British Journal of Psychiatry*, which included Hartigan’s paper. The names were suggested for lithium and imipramine, as the drugs normalizing moods in BD and periodic depression, respectively [7]. Hartigan demonstrated the preventive action against manic and depressive recurrences, which justified naming lithium a drug normalizing (stabilizing) mood. For such drugs, the name “mood stabilizers” has become increasingly used and is now established, whereas in Polish and Russian psychiatric literature, the term “normothymic drugs” has been employed.

In 1967, a paper by Danish psychiatrists already mentioned (Poul Baastrup and Mogens Schou) appeared, summarizing the experiences of 88 patients treated in Glostrup psychiatric hospital with bipolar and unipolar mood disorders, receiving lithium for an average of six years. The authors compared the periods of disturbed mood (mania or depression) per year. The results showed that the mean duration of disturbed mood during lithium administration was more than six-fold shorter than in the period before lithium, indicating a high probability of a favorable prophylactic effect of lithium on the course of mood disorders [8]. However, in the next year, a highly critical article against the Danish research showing a possibility of lithium prophylaxis appeared. It was published in the prestigious journal “Lancet”, authored by the British psychiatrists Barry Blackwell and Michael Shepherd, titled: “Prophylactic lithium: another therapeutic myth?” The authors voiced strong doubts about the results of Danish researchers and postulated performing double-blind studies to verify the issue [9]. As it was in response to the recommendation of the Britons, in 1970–1973, the results of eight placebo-controlled studies carried out in Europe (Denmark and UK) and the USA assessing the prophylactic efficacy of lithium were published. The patients in the trials had at least two episodes of the illness in the last two years. In most studies, the comparison was made between patients in which a placebo replaced lithium and those continuing lithium (so-called “discontinuation design”). The recurrence of the illness was defined as a condition requiring psychiatric hospitalization or antidepressant/antimanic treatment. The overall analysis revealed that the percentage of patients with recurrences of depression or mania was significantly lower during lithium administration (mean 30%) than during placebo (average 70%) [10].

Lithium became an established MS in subsequent years, and its use has significantly increased. At this time, the antidepressant effect of lithium in acute depressive episodes [11,12] and augmentation of antidepressants by lithium [13] were noted. In 1999, marking half a century following lithium’s introduction into contemporary psychiatry [5] and 36 years after Hartigan’s article [4], a Canadian psychiatrist of Czechoslovakian origin, Paul Grof, presented a concept of “excellent lithium responders” (ELRs) for BD patients, in whom lithium monotherapy resulted in total elimination of the illness’ recurrences [14]. At the Poznań center, an assessment of the percentage of ELRs was performed, showing that about one-third of BD patients were free of recurrences during ten years of lithium administration [15]. Several clinical factors have been indicated connected with the good prophylactic efficacy of lithium, such as moderate number of affective episodes with distinct periods of remission, the episode sequence mania–depression–remission, the absence of rapid cycling, and psychiatric comorbidity [16].

In the 21st century, prophylactic lithium efficacy in BD was amply evidenced in three meta-analyses. They showed the superiority of lithium over placebo for preventing all kinds of recurrences, especially manic but also depressive ones [17,18,19]. In 2018, Kessing et al. [20] indicated that lithium monotherapy was prophylactically more effective in the BD population than monotherapy with any other MS. A meta-analysis performed in 2000 confirmed the therapeutic efficacy of lithium in mania [21]. Additionally, lithium augmentation of antidepressants in treatment-resistant depression became an established procedure [22] and was even suggested as the second indication for lithium in psychiatry after BD prophylaxis [16]. Therefore, lithium can meet the most demanding 2 × 2 criteria for MS, indicated by Bauer and Mischner as being therapeutically and prophylactically effective in both mania and depression [23]. Among mood-stabilizing drugs, lithium proved the most efficacious anti-suicidal agent [24]. Lithium also became a unique drug, with which ultra-long-term mood stabilization was experienced. For example, recently, in the Poznań center, we presented the case of a female patient receiving lithium for 50 years, with excellent results in most areas of health and social functioning [25].

### 2.2. Valproate

At the beginning of the 1970s, clinical observations showed that some anticonvulsant drugs may have mood-stabilizing properties. French researchers led by Pierre Lambert, working at Bassens Hospital in Rhône-Alpe, found that valproic acid amide may exert an antimanic and prophylactic effect in BD. In patients with mood disorders, the drug produced a significant fall in manic episodes as well as a decrease in the duration of hospitalizations. They coined, for such a pharmacological activity, the French term “thymorégulatrice”, which can allude to such names as “mood-normalizing” or “normothymic” [26]. Intensive studies on valproates were conducted in the USA in the 1990s. The instrumental researcher in this respect was eminent BD specialist Charles Bowden. Valproic acid amide was not used here, but an equimolar combination of sodium valproate and valproic acid was, named “divalproex”. It transpired that in addition to the solid antimanic activity of this drug, its prophylactic effect preventing BD recurrences was demonstrated, similar to that of lithium [27], paving the way for the wide use of valproates as MSs in treating mania and the prophylaxis of BD.

At the beginning of the 21st century, an increasing use of valproate occurred, partly due to pharmacological companies’ intensive promotion of the drug, with a concomitant decrease in lithium administration. Such a situation imposed the attempts to verify the real efficacy of both drugs, reflected by a project with the acronym BALANCE (Bipolar Affective disorder Lithium/ANtiConvulsant Evaluation). This study assessed the prophylactic efficacy of divalproex monotherapy, lithium monotherapy, and the combination of both drugs for two years in 330 BD patients (110 in each group). Lithium monotherapy had better prophylactic efficacy than divalproex monotherapy, and the best preventive quality for the combination was found [28]. However, a recent paper did not find significant differences between valproate and lithium when used as monotherapy or combined with atypical psychotics in one-year maintenance administration [29].

### 2.3. Carbamazepine

Carbamazepine is another anticonvulsant with potential mood-stabilizing properties. The merit for demonstrating the mood-stabilizing action of this drug should be given to Japanese researchers led by the excellent psychopharmacologist Teruo Okuma (1926–2010). In their 1973 paper, the antimanic effect of carbamazepine was observed in 52% of patients, and the prophylactic activity against manic and depressive episodes was noticed in 74% and 52% of patients, respectively [30]. In 1980–1990, research on the therapeutic and prophylactic effects of carbamazepine in mood disorders was carried out mainly by American psychiatrists under the leadership of Robert Post. The preventive effect of carbamazepine in BD was confirmed [31], as well as the antimanic [32] and possible antidepressant action of the drug [33], which justified treating carbamazepine as a “mood stabilizer”. The drug became popular in Europe, and in the 1990s, carbamazepine became the second medication after lithium used for the prevention of mood disorders’ recurrences. In the late 1990s, German psychiatrists attempted to compare the prophylactic efficacy of carbamazepine and lithium within a study with the acronym MAP (Multicenter study of long-term treatment of Affective or schizoaffective Psychoses), where they assessed efficacy over 2.5 years. They found that lithium was more efficacious in “classic” forms of BD, while carbamazepine more efficacious in atypical ones, e.g., with psychiatric comorbidity or mood-incongruent delusions [34].

The introduction, in the 1990s, of the carbamazepine derivative oxcarbazepine into psychiatric treatment should also be mentioned. The antimanic and prophylactic efficacy of oxcarbazepine in BD was demonstrated, indicating that some pharmacokinetic and side-effect properties of this drug can be more favorable than those of carbamazepine [35].

## 3. Second Generation of Mood Stabilizers (SGMSs)

### 3.1. Clozapine

More than three decades passed from Hartigan’s paper when American psychiatrists observing the effects of clozapine, the drug introduced to the USA several years earlier, suggested that it can possess mood-stabilizing properties. In 1995, a paper appeared with Carlos Zarate as the first author [3], where 17 patients receiving clozapine for the treatment of mania successfully were followed up for 16 ± 6 months. Eleven of them (65%), during the administration of clozapine, had no recurrences or hospitalizations. Previously, the antimanic effect of clozapine had been widely observed. In the Poznan center, such an effect had been noted in the early 1980s [36]. A systematic review and meta-analysis performed in 2020 found the therapeutic effect of clozapine in mania equal to other antipsychotics and even better among treatment-resistant patients [37]. Employing clozapine in treatment-resistant mania was also suggested by major bipolar disorder specialists [38].

Thus, clozapine initiated the second generation of mood stabilizers (SGMSs) [2]. In 2015, Chinese authors performed a meta-analysis on long-term treatment with clozapine, where fifteen papers were covered, including more than 1000 patients. In the article, the long-term efficacy of clozapine, both therapeutic and prophylactic, was confirmed. It was shown that the drug used either as monotherapy or combined with other MSs produced a significant amelioration in mania, depression, psychosis, and rapid cycling. A substantial number of the patients significantly improved or achieved remission [39].

Therefore, clozapine can be helpful for preventing recurrences in severe and drug-resistant BD. In the recent Polish standards, the longitudinal administration of clozapine is suggested for treatment-resistant bipolar patients, providing sufficient hematological monitoring. Combination with lithium can augment its efficacy and diminish the risk of leukopenia. The predictive factor for the good prophylactic efficacy of clozapine in bipolar affective disorder is severe manic episodes with psychotic symptoms and substantial agitation [40].

### 3.2. Olanzapine

The next atypical antipsychotic meeting the above criteria for MS is olanzapine, which is structurally related to clozapine. Olanzapine, in doses of 10–20 mg/day, demonstrated excellent clinical efficacy and good tolerance in treating mania [41]. According to a comparative analysis, olanzapine exhibits the best profile of efficacy and tolerability among antimanic drugs [42]. In bipolar depression, olanzapine exerted a significant therapeutic effect when combined with fluoxetine. The effectiveness of composite olanzapine–fluoxetine treatment of drug-resistant depression and psychotic depression in bipolar and unipolar mood disorder has also been demonstrated [43]. Olanzapine can also serve to augment antidepressant drugs in treatment-resistant depression [44].

Olanzapine monotherapy for the prevention of recurrences in BD was compared to placebo and first-generation mood stabilizers in multiple controlled studies. The high prophylactic efficacy of olanzapine monotherapy was confirmed, mostly against manic episodes and especially in patients in whom the drug was therapeutically effective during such an acute episode [45]. When comparing olanzapine with lithium, it was found that mania recurrences were notably less frequent with olanzapine [46]. Polish recommendations suggest olanzapine monotherapy as the first-line prophylactic treatment in type I BD showing a preponderance of manic episodes [40].

### 3.3. Quetiapine

Quetiapine, similar to lithium, meets the stringent criteria for an MS to be therapeutically and prophylactically effective in both psychopathological poles [24,47]. In the manic state, quetiapine monotherapy is effective at 400–800 mg/day doses, and its efficacy is similar to that of lithium [48]. In placebo-controlled studies, quetiapine, at an amount of 300 or 600 mg/day, was found to be effective in treating depression in the course of bipolar affective disorder type I or type II. Such an efficacy of quetiapine was shown to be even more significant than that of lithium [49]. Quetiapine monotherapy has been proposed for treating bipolar depression in most guidance [50]. The drug can also be used for enhancing the antidepressant medications’ efficacy in both bipolar and unipolar depression resistant to treatment [44].

The antipsychotics of the second generation primarily prevent manic recurrences, whereas quetiapine monotherapy is effective in preventing both manic and depressive episodes. For depressive recurrences, the drug’s monotherapy has been equally efficacious as that of lithium [51]. A four-year comparison of the prophylactic efficacy of quetiapine in monotherapy or combination with other MSs showed that the percentage of patients without recurrences on quetiapine monotherapy was 29.3%. However, when combining quetiapine with lithium, no recurrences occurred in 80%, and with valproate, in 78.3% of patients [52]. In Polish standards, quetiapine monotherapy is among the first-line treatments for long-term therapy in bipolar disorder with a predominance of depressive episodes [40].

### 3.4. Aripiprazole

Aripiprazole is an atypical antipsychotic that started the third generation of these drugs [53]. It joined the family of MSs meeting the criteria above in 2007, when a prophylactic effect of the drug’s monotherapy was demonstrated in a two-year study. In this, aripiprazole significantly prevented the recurrences of manic episodes but not the depressive ones [54]. In a recent study, such prevention was also demonstrated for long-acting injectable (LAI) aripiprazole [55]. Previously, aripiprazole at doses of 15–30 mg/day was found to exert a significant anti-manic effect [56]. Further studies showed that such therapeutic efficacy was similar to lithium [57]. Aripiprazole in smaller doses (5–10 mg/day) has been used to enhance antidepressants’ effectiveness in case of their suboptimal outcome and has been recommended for the augmentation of antidepressants in drug-resistant depression [58].

### 3.5. Risperidone

Risperidone qualified as an SGMS in 2010 when a two-year study indicated that the drug’s monotherapy in the form of LAI showed a significant prophylactic effect concerning recurrences of manic episodes in patients with type I bipolar affective disorder [59]. Such an impact of LAI risperidone was also confirmed in more recent observations [55]. Previously, it was found that risperidone at doses of 1–6 (average 3–4) mg/day was therapeutically effective in mania [60] and presented, second to olanzapine, the best efficacy and tolerance profile in the treatment of manic episodes [42]. Risperidone does not exert an antidepressant effect. However, low doses of the drug can be used to augment antidepressants in treatment-resistant depression [44].

### 3.6. Lamotrigine

In addition to atypical antipsychotic drugs, the SGMSs were supplemented by a new anticonvulsant drug, lamotrigine. Among all mood-stabilizing drugs, lamotrigine exerts the most potent antidepressant activity and is, thus, regarded as a “mood stabilizer from below”. Lamotrigine is effective in the treatment of bipolar depression [61]. The favorable effects of lamotrigine were also described in brief recurrent depression [62]. In the prophylaxis of BD, the drug prevents mainly depressive recurrences. In a 1.5-year comparative study of lithium and lamotrigine, lithium was significantly better in preventing manic episodes. In contrast, lamotrigine was superior in the prevention of depressive ones [63]. Promising effects of lamotrigine were also reported in the rapid cycling BD [64]. Predictive factors for the prophylactic efficacy of lamotrigine are different from those for lithium and clozapine. A good prophylactic effect of lamotrigine can be obtained in patients with chronic depressive episodes, rapid cycling, and comorbid anxiety conditions, e.g., panic disorder [65].

### 3.7. Lurasidone

In recent years, a candidate for an MS was a second-generation antipsychotic, lurasidone, on account of its therapeutic activity in bipolar depression and prophylactic effects in BD. In bipolar depression, lurasidone, employed as monotherapy or in combination with lithium or valproic acid, produced a substantial decrease in depressive symptoms [66], corroborated by meta-analyses and systematic reviews [67]. Furthermore, a trial of lurasidone, 20 to 120 mg daily, for up to two years, following its use in bipolar depression, in a large group of patients, showed its safety and prophylactic efficacy as monotherapy and in combination with lithium or valproate [68]. In a one-year study in Japan, including BD patients, the drug was used in the same doses as above, with or without lithium or valproate. It was shown that in previously treated patients with bipolar depression, the drug continued the betterment in depressive symptoms and also improved psychiatric conditions in patients with manic, hypomanic, or mixed episodes [69]. Therefore, it seems that lurasidone may successfully meet the criteria for becoming an SGMS.

## 4. “Insufficient” Mood Stabilizers

Several drugs have shown some therapeutic and prophylactic efficacy in bipolar disorder. However, they do not sufficiently meet the criteria for the MSs listed at the beginning of the paper. Nevertheless, they may be helpful mainly as an add-on to established MSs to augment their efficacy and treat psychiatric and somatic comorbidity.

### 4.1. Atypical Antipsychotics

#### 4.1.1. Asenapine

Asenapine was found to be therapeutically effective in mania and probably depression. At doses of 10–20 mg/day, it showed good efficacy and tolerability in the treatment of mania, which was confirmed in a meta-analysis published in 2013 [70]. In the recent recommendations of the Canadian Network for Mood and Anxiety Treatments (CANMAT), asenapine was regarded as among the most valuable medications for treating mania [50]. In patients with BD type I treated for manic or mixed episodes, a post hoc analysis revealed a reduction by asenapine in depressive symptoms [71]. A therapeutic effect of the drug was also shown in some bipolar depressed patients [72]. Szegedi et al. [73] investigated the prophylactic efficacy of asenapine in patients with BD type I in a randomized, placebo-controlled study. They observed that asenapine prevented both manic and depressive episodes when started after the acute ones [73]. However, the trial lasted only half a year, which was insufficient to meet the second criterion for MSs, requiring at least one year of study [1].

#### 4.1.2. Ziprasidone

Ziprasidone was found to be efficacious and had good tolerability in mania [74]. However, there is no study on ziprasidone monotherapy for the maintenance treatment of BD. The addition of ziprasidone to monotherapy with lithium or valproate improved the efficacy during a 6-month follow-up [75].

#### 4.1.3. Paliperidone

In a randomized, double-blind study published in 2010, paliperidone extended-release (ER), 12 mg/day, was superior to a placebo in the treatment of mania [76]. Two years later, a paper presented the results of a long-term placebo-controlled study. It showed that in patients with BD type I, paliperidone ER prevented manic but not depressive episodes when initiated after an acute manic or mixed one [77]. However, the responder-enriched design of this study does not allow us to infer the conclusion for the whole group of bipolar patients.

#### 4.1.4. Cariprazine

Cariprazine, another third-generation antipsychotic, was therapeutically effective in mania and bipolar depression. In the treatment of acute and mixed mania, cariprazine, both in low (3–6 mg/day) and high (6–12 mg/day) doses, was found to be efficacious in a double-blind, placebo-controlled study [78]. The drug also proved to be effective and well-tolerated in bipolar depression when used at 1.5–3 mg/day in a double-blind placebo-controlled trial. Based on them, in 2019, the Food and Drug Administration approved cariprazine for treating bipolar depression [79]. However, there have been no results on cariprazine monotherapy in the maintenance treatment of BD.

#### 4.1.5. Brexpiprazole

The successor of aripiprazole, brexpiprazole, was tested in a placebo-controlled study for the acute treatment of bipolar mania. In the short-term treatment, there was no difference between brexiprazole and placebo; however, the open-label extension demonstrated a gradual improvement in manic symptoms with the drug [80]. Another pilot study found some improvement after brexpiprazole treatment in bipolar depression [81]. No maintenance trial of brexpiprazole in BD has been performed so far.

#### 4.1.6. Lumatoperone

Lumateperone, at 42 mg/day, significantly improved depression symptoms in patients with depression in the course of BD, both type I and II [82], whereas no trials with lumateperone were performed in mania and maintenance treatment of BD.

### 4.2. Anticonvulsants

#### 4.2.1. Topiramate

Anticonvulsant topiramate seemed a promising drug for mood-stabilizing agents when reports of its antimanic action used as monotherapy or add-on treatment appeared [83,84]. However, Cochrane analysis in 2016 did not fully confirm these claims [85]. Apart from positive case reports [86], there were no long-term trials with topiramate monotherapy in the maintenance treatment of BD, whereas, in a one-year study, adjunctive topiramate significantly reduced new manic and depressive episodes [87]. However, despite not fulfilling the criteria for MSs, adjunctive topiramate could be helpful in some special conditions in BD. The first is a reduction in weight gain. A favorable effect in this respect was already evidenced in a 24-week study [88]. Topiramate also demonstrated therapeutic properties in alcohol addiction [89] and some anxiety disorders [90]. However, no drug trials in BD patients with these comorbidities have been performed so far.

#### 4.2.2. Gabapentin

The attempts to use gabapentin in BD mostly concerned add-on therapy. In 1999, the possible effectiveness of adjunctive gabapentin was shown in patients with drug-resistant bipolar states, particularly concerning depressive symptomatology [91]. Several years later, the same authors, showing such an effect on a higher number of patients, suggested that the drug’s effectiveness may be due to a simultaneous impact on anxiety and alcohol abuse comorbidity [92]. In a one-year double-blind placebo-controlled study, Vieta et al. [93], demonstrated the prophylactic efficacy of adjunctive gabapentin with no emerging manic and depressive symptoms. Thus, although the recent meta-analyses question the effectiveness of gabapentin in BD [94,95], it seems that the adjunctive use of the drug may augment the therapeutic and prophylactic efficacy of the well-established MSs, probably mostly in patients with concomitant anxiety and alcohol misuse.

#### 4.2.3. Pregabalin

Pregabalin is an active metabolite of gabapentin, and its introduction into psychiatry was connected to its therapeutic effect on generalized anxiety disorder. The pursuits of using the drug in BD are fewer than those of gabapentin and pertain primarily to adjunctive therapy. In 2009, a case report described the successful use of pregabalin as an add-on to quetiapine in treating acute mania [96]. In 2013, the promising results of an open trial of pregabalin as an acute and maintenance adjunctive treatment for 58 outpatients with treatment-resistant bipolar disorder were presented. There were neither serious side effects nor adverse drug interactions with other MSs [97], although, similar to gabapentin, pregabalin does not meet the criteria for an MS. However, its adjunctive use in the treatment and prophylaxis of BD can be considered, especially with patients with generalized anxiety comorbidity.

### 4.3. Memantine

Memantine is a pro-cognitive drug acting mainly through a glutamatergic NMDA receptor. Studies in recent decades have shown that the drug may also make an application as an adjunctive agent in the treatment and prophylaxis of mood disorders. Italian investigators in a one-year open trial of 40 treatment-resistant BD patients showed that adjunctive memantine was associated with a clinically substantial antimanic and mood-stabilizing effect, with an excellent safety and tolerability profile [98]. They further demonstrated that such an effect was sustained after three years [99]. In a recent case report, an antidepressant effect of adjunctive memantine in bipolar patients was described [100]. Therefore, like the anticonvulsant drugs above, memantine does not meet the criteria for an MS as monotherapy. However, its adjunctive use in the treatment and prophylaxis of BD can be considered. It may be hypothesized that it can be of particular benefit to patients with cognitive problems.

## 5. Preventing Recurrences of Mood Episodes Is the Core Aspect of Mood Stabilizers

Recently, Brazilian investigators presented a theoretical mood stabilizer model based on its definitions in the relevant literature. Their concept analysis unearthed four attributes of an MS, such as “not worsening”, “acute effects”, “prophylactic effect”, and “advanced effect” [101]. The first three attributes were adequately covered by the author’s definition presented at the beginning of this paper. Additionally, the author agreed that the prophylactic effect can be considered the core aspect of a legitimate MS. Furthermore, in the author’s requirements for the prevention of manic and/or depressive recurrences during long-term administration, a drug should be given as monotherapy and a trial performed for at least one year. The “advanced effect” may involve additional benefits from prophylactic treatment; however, such a term is poorly defined [102].

Therefore, the most essential aspect of an MS is preventing affective episodes’ recurrences. In this regard, sixty years of clinical experience have shown the unquestionable efficacy of mood-stabilizing drugs for such prevention. Bipolar patients in whom lithium monotherapy results in the total elimination of recurrences constitute about 30% [15]. A similar percentage of excellent prophylactic efficacy may be obtained with other MSs, although clinical characteristics of such responders could be specific for each drug, an example of which can be the features of a good responder to clozapine [40], olanzapine [45], or lamotrigine [65]. Therefore, it transpires that in the majority of BD patients, for obtaining an optimal prophylactic effect, combination therapy with mood-stabilizing drugs is needed. The combination of atypical antipsychotic drugs with lithium or valproate has shown a better prophylactic effect than monotherapy with each of these drugs [103,104,105], as was the combination of lithium and valproate [28].

Japanese researchers [106] performed a meta-analysis of the prophylactic efficacy of many configurations of FGMSs and SGMSs in BD. In their examination, aripiprazole, asenapine, olanzapine, paliperidone, and risperidone LAI were better than placebo for recurrence prevention of any mood episode. A combination of antipsychotics with lithium or valproate (LIT/VAL) was better than LIT/VAL placebo (except for olanzapine) for such a prevention. Both lurasidone and quetiapine + LIT/VAL were better than the LIT/VAL placebo in the prevention of depression. On the other hand, both aripiprazole and quetiapine + LIT/VAL outperformed the LIT/VAL placebo in the prevention of mania, whereas both lurasidone and quetiapine + LIT/VAL surpassed the LIT/VAL placebo in the discontinuation for all causes [103].

Because of a new classification of psychotropic drugs known as the Neuroscience-based Nomenclature (NbN) [107], it could be interesting to note how the FGMSs and SGMSs fit into the NbN. Out of the FGMSs, lithium is depicted as an enzyme modulator and valproates and carbamazepine as channel blockers. The latter is also applied to the SGMS lamotrigine and to “infufficient” ones, such as gabapentin and pregabalin. Most SGMSs (clozapine, olanzapine, risperidone, lurasidone) are defined as antagonists of dopamine and serotonin systems, along with such “insufficient” ones as asenapine, paliperidone, and ziprasidone. Aripiprazole SGMS is defined as a partial agonist and antagonist of the dopaminergic and serotonergic system, similar to brexpiprazole and cariprazine, having the status of “insufficient”. Quetiapine SGMS is defined as multimodal, similar to lumateperone, which has the status of insufficient MS. Finally, two insufficient MSs, topiramate and memantine, are defined as mainly acting on the glutamatergic system, with memantine being the NMDA antagonist.

## 6. Suggestions for the Long-Term Use of Mood Stabilizers

Polish recommendations for the long-term use of MSs in BD were elaborated by the author of this review in the recent edition of standards for pharmacological treatment, published in 2022. Perhaps they are not be applicable internationally. However, most of them are similar to the CANMAT [50] as well as to the guidelines of CINP [108] and WFSBP [109]. The Polish suggestions cover both monotherapy and combinations for type I and II of BD. Lithium is the drug of choice in the “classic” form of BD I, with moderate frequency or episodes, distinct periods of remission, and the absence of significant central nervous system pathology. In a suboptimal effect after a year, adding a second MS of the first or second generation is suggested. In an illness with dominant manic episodes, olanzapine is recommended, especially after an excellent therapeutic effect in such an episode. In a low impact after a year, we propose a combination with one of the FGMSs. If side effects of olanzapine appear (e.g., excessive weight gain), the drug can be changed to quetiapine or aripiprazole. In BD type I with dominant depressive episodes, the administration of lamotrigine or quetiapine is recommended. In a suboptimal effect, a combination of lamotrigine or quetiapine with an FGMS is suggested, preferably with lithium or a combination of lamotrigine with quetiapine. In an illness with atypical features (mixed episodes, mood-incongruent delusions, comorbidity of anxiety disorders), anticonvulsant drugs (valproates, carbamazepine, or lamotrigine) are indicated as the first choice. In the case of insufficient effect, lithium or mood-stabilizing antipsychotics could be added. In treatment-resistant BD I, the drug of choice is clozapine as monotherapy, with an option of adding an FGMS, preferably lithium, which can prevent leukopenia. In the rapid-cycling form, the combination should be introduced from the start, beginning with two FGMSs, and an SGMS can be later added as a third drug. Lithium is also the drug of choice in the classic type of BD type II, which can be further combined with another FGMS. In the majority of such patients, there is a possibility of long-term administration of antidepressant drugs. Similar to BD I, in the atypical form of BD II (e.g., pathological EEG), initial recommendations point to carbamazepine or valproates, and after a suboptimal effect, lithium can be added. In many such patients, antidepressant drugs can be used. Promising experiences have been obtained for lamotrigine monotherapy in BD II with rapid cycling [60]. After a suboptimal effect, an FGMS can be added, preferably lithium, and next, an atypical antipsychotic as the third drug. In this form, antidepressant medications should be avoided [40].

A drug choice could also be based on a patient’s clinical characteristics. The use of lithium should be considered in each patient with a high risk of suicide. A psychiatrist can assess such risk factors based, among others, on a family history of suicide, previous suicidal behavior in the patient, and his/her present life situation and clinical status. Therefore, lithium should constitute an element of combination therapy or monotherapy if the patient shows clinical characteristics of ELR. When structural changes in the brain are discovered (magnetic resonance, computer tomography, or abnormal EEG), anticonvulsant drugs of the first generation (valproates, carbamazepine, or oxcarbazepine) or second generation (lamotrigine) are indicated. In the presence of mixed manic states, the anticonvulsants (valproate, carbamazepine) or mood-stabilizing antipsychotics (olanzapine, clozapine, quetiapine) show the best effect. In depressive mixed conditions, lithium, lamotrigine, or quetiapine are indicated as monotherapy or in combination. In comorbid anxiety disorders, lamotrigine should be considered, or a combination of an MS with gabapentin or pregabalin. In patients abusing alcohol or psychoactive substances, anticonvulsant drugs of the first or second generation as well as some atypical antipsychotics (clozapine, quetiapine) could be recommended, as well as adjunctive gabapentin. A combination with memantine could be considered in the case of cognitive problems. For body mass, among the FGMSs, the most neutral is carbamazepine and the SGMSs aripiprazole and lamotrigine,

In summary, the prudent use of the FGMSs and SGMSs could provide the most efficient long-term management of patients with bipolar mood disorders.

## Data Availability

Not applicable.

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
