# Peer review of "Mood Stabilizers of First and Second Generation"

_brainsci, 2023, doi:10.3390/brainsci13050741_

Round 1

Reviewer 1 Report

This paper provides a pseudo historical overview and justification for the use of various medications for the treatment of BPD currently accepted internationally for the disorder. There is an emphasis on Polish guidelines for the disorder which may not be applicable internationally. For example, the use of clozapine as advocated here would not be acceptable in many countries (reserved for treatment resistant schizophrenia only) although the author makes a case for less restrictive use. Perhaps with respect to guidelines the author could provide a brief comparative table of Polish versus CANMAT versus CINP versus WFSBP recommendations as first and second line treatment for mania, BP depression and prophylaxis?

I am not convinced of the authors argument that valproate advertising by pharm caused the decline in lithium use. While it may have contributed it seems to me that an equally plausible explanation is patient preference, based on the factors that lithium 1: is not well tolerated by many patients; 2: its effects on renal and thyroid function that may require cessation of treatment; 3: its necessity for plasma monitoring which patients may also find inconvenient.

There could be some tightening of the English usage but as currently written the meaning is not obscured.  

The number system needs attention.

A native speaker of English may have used some different language construction for example line 46: "we observe the sixtieth anniversary......."

Author Response

This paper provides a pseudo historical overview and justification for the use of various medications for the treatment of BPD currently accepted internationally for the disorder. There is an emphasis on Polish guidelines for the disorder which may not be applicable internationally. For example, the use of clozapine as advocated here would not be acceptable in many countries (reserved for treatment resistant schizophrenia only) although the author makes a case for less restrictive use. Perhaps with respect to guidelines the author could provide a brief comparative table of Polish versus CANMAT versus CINP versus WFSBP recommendations as first and second line treatment for mania, BP depression and prophylaxis?

Response: In the title of chapter 6, the term "recommendation" was changed on "suggestions". Also, the following was added: Perhaps they may not be applicable internationally. However, most of them is similar to the CANMAT [50] as well as to the guidelines of CINP [108] and WFSBP [109] (lines 456-8).

I am not convinced of the authors argument that valproate advertising by pharm caused the decline in lithium use. While it may have contributed it seems to me that an equally plausible explanation is patient preference, based on the factors that lithium 1: is not well tolerated by many patients; 2: its effects on renal and thyroid function that may require cessation of treatment; 3: its necessity for plasma monitoring which patients may also find inconvenient.

Response: The sentence was rephrased (lines 137-9)

There could be some tightening of the English usage but as currently written the meaning is not obscured. A native speaker of English may have used some different language construction for example line 46: "we observe the sixtieth anniversary......."

Response: The sentence was changed into: "Sixty years ago, the article appeared..." (line 46)

The number system needs attention.

Response: This was corrected.

Reviewer 2 Report

General remarks

The paper is focused on practical issues, very didactic, understandable , nice written.  The paper is mainly aimed to clinically oriented psychiatrists. To take into consideration aims and scope of journal (a scientific journal that publishes original articles, critical reviews, research notes, and short communications in the broad area of neuroscience) a new classification of psychotropics the Neuroscience-based Nomenclature  (NbN) should be mentioned.  NbN is a publication and a digital application of psychiatric medications classified by their pharmacology and mode of action. The core principles of the NbN are to move from a disease-based classification (e.g., antidepressants, antipsychotics, anxiolytics, stimulants and mood stabilisers) to a pharmacologically driven classification, in order to shift from symptoms to mechanisms, from disease to pharmacology. Please add a commentary to this including the  perspective of psychopharmacology  (personalized treatment, precision psychiatry medicine).

Concrete remarks

Page 6, line 282

Lurasidone (SDA) belongs to the 2nd generation antipsychotic -please correct.

Ad section 4 (Insufficient MS)

Brexpiprazole and lumateperone should be mentioned too (in agreement to the definition for insufficient MS)

Author Response

The paper is focused on practical issues, very didactic, understandable, nice written.  The paper is mainly aimed to clinically oriented psychiatrists. To take into consideration aims and scope of journal (a scientific journal that publishes original articles, critical reviews, research notes, and short communications in the broad area of neuroscience) a new classification of psychotropics the Neuroscience-based Nomenclature  (NbN) should be mentioned.  NbN is a publication and a digital application of psychiatric medications classified by their pharmacology and mode of action. The core principles of the NbN are to move from a disease-based classification (e.g., antidepressants, antipsychotics, anxiolytics, stimulants and mood stabilisers) to a pharmacologically driven classification, in order to shift from symptoms to mechanisms, from disease to pharmacology. Please add a commentary to this including the  perspective of psychopharmacology  (personalized treatment, precision psychiatry medicine)

Response: The NbN was mentioned and all mood stabilizers were also defined according to this classification (lines 439-451)

Concrete remarks

Page 6, line 282

Lurasidone (SDA) belongs to the 2nd generation antipsychotic -please correct.

Response: This was corrected

Ad section 4 (Insufficient MS)

Brexpiprazole and lumateperone should be mentioned too (in agreement to the definition for insufficient MS)

Response: Both drugs were added to section 4, as 4.1.4 and 4.1.5 (lines 339-350)